# Does More Managerial Power Impede or Promote Corporate Tax Avoidance? Evidence from Listed Chinese Companies

**Yingkai Tang [1], Yao Liu [1], Jing Liu [1] and Weiping Li [2],\*** 

[1] Business School, Sichuan University, Chengdu 610065, China; tang@china.com (Y.T.); liuyao183@gmail.com (Y.L.); liujing@scu.edu.cn (J.L.)

[2] Department of Economics and Finance, City University of Hong Kong, Hong Kong 999077, China

\* Correspondence: liweiping160@gmail.com

**Abstract:** Paying taxes is a form of corporate social responsibility. Corporate tax avoidance effectively reduces operating costs, thereby increasing shareholder returns, but the pros and cons are different for executives. This paper takes data from companies listed in China from 2000 to 2016 to explore the causal relationship between managerial power and corporate tax avoidance, using principal component analysis and entropy weight methodology to construct managerial power. First, we conduct an ordinary least square regression, and then we employ alternative measures to avoid taxation, and use managerial power as a robustness check. Next, we rerun the model by using quantile regression and propensity score matching. Finally, in order to tackle problems with endogeneity, we carry out regressions utilizing instrumental variables and simultaneous regression equations. We conclude that managerial power reduces corporate tax avoidance, and firms with more managerial power have fewer incentives to avoid taxes. The results of this paper can guide the taxation of companies listed on the Chinese market to achieve the full payment of tax obligations by reinforcing managerial power in companies, which improves national taxation and leads to sustained economic growth.

**Keywords:** managerial power; tax avoidance; instrument variable; propensity score matching

## 1. Introduction

Corporate taxation contributes to social welfare and is a corporate social responsibility (CSR). According to CSR, companies may use resources for socially responsible activities that do not necessarily benefit them. Many scholars have recognized the complementary relationship between CSR and taxation behavior [1–5]. They believed that a company's goal is not only to maximize profits, but also to consider the interests of all stakeholders [1,2], which encourages companies to engage in socially responsibility activities. Paying taxes is the most basic way for private and corporate citizens to participate in a socially responsible activity [3].

Corporate tax avoidance refers to the reasonable and legal use of tax law imperfections and tax incentives to evade or reduce tax liabilities [4]. Recent research has found that tax avoidance moderated the effect of institutional ownership, and that firm size affected firm value [5]. Klepper [6] stated that "three things in life are certain: death, taxation, and the unremitting efforts of human beings to escape these two things." Taxation, as one of the most important sources of fiscal revenue, has an essential role in regulating the allocation of economic and social resources. Tax avoidance behavior leads to a large reduction in the state's fiscal revenue that adversely affects the country's welfare policy, making the government unable to provide good social public services, disrupting the

normal social and economic order, and destroying market resources. Tax avoidance is a common tax planning strategy in companies, although there are vast differences in degree across companies. For these reasons, exploring tax avoidance has become a hot topic in academic circles.

Scholars have studied the factors that influence tax avoidance from different angles. Studies have found that corporate size, profitability, debt levels, and political connections [7,8], as well as tax administration, external audits, external professional investors, and environmental factors [9] all affect corporate tax avoidance. Similarly, many scholars also believe that senior management, as corporate tax policy decision makers, have a large impact on corporate tax avoidance, but scholars disagree about their precise impact. For example, Phillips [10] believed that when companies use performance as the basis for executive incentives, executives are more inclined to conduct corporate tax avoidance to improve the company's earnings and meet their own interests. However, Desai [11] disagreed with those scholars. He thought that in companies with relatively poor governance mechanisms, rent seeking due to managerial power means that management incentives reduce corporate tax avoidance behavior.

These inconsistent scholarly views became the direct reason for this paper. As far as the Chinese market is concerned, its special corporate governance environment is also in urgent need of special study. In general, there is a widespread lack of investor protection in Chinese companies, which leads corporate executives to use their authority for self-interest. In addition, China's enterprise system has major defects, such as unclear management levels, especially in state-owned enterprises. The special circumstances of the Chinese market mean that foreign research is not relevant in China; this is especially true because the existing research holds different, and even contrary, views on corporate executives' tax avoidance behavior.

This paper combines managerial power with corporate tax avoidance using data from companies listed in the China market from 2000 to 2016, constructing managerial power through principal component analysis (PCA) and entropy weight methods. It then explores the role and path of managerial power on corporate tax avoidance through ordinary least square (OLS) regression, quantile regression, propensity score matching (PSM), instrumental variable regression, and other methods to provide a reference for Chinese companies.

Overall, this study does some contributions to theoretical research and practice. In terms of theoretical research, it enriches the study of managerial power and corporate tax avoidance, and may provide a new possible research perspective for corporate tax avoidance. The practical finding of this study is that managerial power inhibits corporate tax avoidance. Therefore, when conducting business guidance, government departments should implement more tax incentives by increasing managerial power, thus improving national taxation and economic growth.

The rest of the paper is organized as follows: Section 2 presents a literature review and research hypotheses. Section 3 explains the data and methodology. Section 4 provides the empirical results and performs some robustness tests. Finally, Section 5 delivers the conclusion.

## 2. Literature Review and Hypotheses

### 2.1. The Literature on Corporate Tax Avoidance

As a means of saving on taxes and retaining profits, corporate tax avoidance is highly correlated with corporate value [11]. Therefore, exploring which factors affect companies' tax avoidance behavior has great practical value. In recent years, many scholars have studied the factors that influence tax avoidance. In the following, we divided scholars' views into two categories: impeding or promoting tax avoidance.

From the impediment viewpoint, Chen et al. [12] thought that a higher equity concentration led to stronger risk aversion by the controlling shareholder and a greater inclination to adopt a positive tax avoidance policy, especially in family-owned businesses. Gaertner [13] believed that shareholders and investors needed to change the tax policies of risk-averse executives through appropriate incentives.

If executives received fewer monetary gains from explicit pay contracts, they would undertake more self-interested transactions, such as rent seeking.

Austin and Wilson [14] believed that social awareness would increase tax avoidance costs, thus impeding tax avoidance. Similarly, Dowling [15] reasoned that media attention would impede corporate tax avoidance by increasing the probability of exposing tax incentives, bringing higher costs, and strengthening the shareholders' supervision of management. Even a low level of corporate governance could reduce firm tax avoidance.

In addition, improving the financial development environment [16] and creating a more reasonable board system [17] could also effectively impede the emergence of corporate tax avoidance.

As for the promotion viewpoint, this research has focused mostly on executives. As Slemrod [18] pointed out, since tax avoidance was a valuable act that could bring economic benefits to shareholders, then shareholders should create incentives to ensure that managers make decisions that avoid taxes. Moreover, according to principal-agent theory, corporate tax avoidance could become a tool for managers' self-interested behavior. Zhou and Li [19] discovered a clear positive correlation between relational transactions and corporate tax avoidance, and that female managers weakened the positive correlation and promoted relational transactions, triggering more aggressive tax avoidance activities. Dong and Hu [20] found a positive correlation between the amount of executive incentives and the level of tax avoidance. Hsieh et al. [21] observed that firms with overconfident CEOs and CFOs were more likely to participate in tax avoidance activities. Reputation was another motivation for executives to use power to avoid tax, consistent with Bornemann's [22] research.

Executive compensation was one of the most important factors that promoted corporate tax avoidance. In China, in-service consumption was an important part of managers' implicit remuneration, and tax avoidance provided a convenient way for them to hide it. Therefore, Luo and Zeng [23] pointed out that the greater the salary competition for managers, the greater the likelihood that managers would use tax avoidance to obtain hidden benefits. Zhang Xinmei [24] believed that executives with a CPA background would be more capable at corporate taxation strategies than non-CPA executives.

We saw that scholars had varying opinions on the factors affecting corporate tax avoidance. At the same time, we realized that scholars have not yet studied the effect of managerial power on tax avoidance. It is clear that the managerial power of its senior executives, the ones who make the decisions on tax avoidance, directly affects a company's degree of tax avoidance, but how it does so, and the extent of the impact, is a topic worth studying.

*2.2. The Literature on Managerial Power*

The senior executives of a company possess the actual managerial power, and thus they effectively control the decision-making of the company. Part of senior executives' managerial power comes from their shareholding ratio. La Porta et al. [25] studied the relationship between corporate financial fraud and shareholding structure, and discovered that higher equity concentration led to poorer financial reporting. They also found that, when director equity reached a certain level, it granted managers control over their positions, at which point the managers used that power for their own interests. Fan et al. [26] discovered that there was no single linear relationship between executive shareholding ratio and the degree of enterprise tax avoidance. When the executive shareholding ratio of was low, managers were subject to equity incentives, their interests were in line with ownership's interests, and the value of the company increased. However, when the executive shareholding ratio was too high, the effect of equity incentives declined, and managers increased their rent-seeking behavior to maximize their own interests. At the same time, they pointed out that the higher the concentration of the listed company's equity, the more serious the agency problem became, and the more likely the company was to commit violations. Erickson et al. [27] learned that the greater the executive shareholding ratio, the higher the level of corporate financial fraud and the greater the degree of tax avoidance.

Aggarwal et al. [28] observed that executives' pay sensitivity was closely related to their responsibilities and power. As their responsibilities grew, executives' compensation performance became more sensitive. In China, however, this sensitivity had different consequences. Firth et al. [29] found a positive relationship between salary performance and pay sensitivity in China, but the relationship was not statistically significant.

In addition, managerial power greatly affected companies' operating atmosphere. The American Association of Fraud Examiners pointed out that employees were very sensitive to the behavioral tendencies of executives, and based their actions on executive attitudes. By pursuing gains at the cost of skirting the edge of the law, employees used any available means to increase the company's wealth.

At the same time, managerial power had a large impact on the financial management of enterprises. Dechow et al. [30] investigated the SEC Enforcement Actions for violations of General Accounting Standards, and found that if the CEO was the founder of the company or on the board, earnings manipulation was more common. D'Aquila [31] confirmed that high-level voices had a big impact on the authenticity of a company's financial reports, and were a manifestation of managerial power. Hunton et al. [32] conducted a two-stage regression study of the impact of high-level voices on a company's earnings quality, confirming that high-level voices were directly proportional to a company's earnings quality. They also found that factors such as board quality, executive age, and executive incentives affected the establishment of high-level voices. The board of directors also indirectly influenced high-level voices, generally via the CEO. Dyreng et al. [33] believed that high-level voices affected corporate tax avoidance behavior in two ways: on the one hand, executives used their knowledge of the industry and the company's characteristics to adjust the distribution of company resources. On the other hand, company executives provided incentives to tax executives to make decisions on taxes that were consistent with the executives' preferences.

Bertrand and Schoar [34] explicitly introduced the power of individual managers into the study of corporate behavior for the first time, confirming that, after controlling for other variables, the power of individual managers affected a company's decision-making behavior and performance. Following in Bertrand and Schoar's footsteps, some scholars have found that executives have a big impact on financial disclosures [35]. Dyreng et al. [33] controlled for other variables that influenced corporate tax avoidance and found that changes in corporate leadership led to significant changes in corporate effective tax rates.

In summary, we can see that, although the research on the operation and financial quality of companies by executives is adequate, research focused on the role of managerial power in corporate tax avoidance behavior is lacking. In addition, most of the studies above examined capital markets outside China; research on the Chinese market is even weaker. Therefore, our research on managerial power and corporate tax avoidance has great theoretical and practical importance.

### 2.3. Hypothesis Development

The literature review shows that different researchers have different views on the relationship between managerial power and tax avoidance. Some scholars thought that rent-seeking behavior [11], high society awareness [14], and media attention [15] impeded tax avoidance. However, other scholars believed that high incentives [8], executives' self-interested behavior [18], invisible in-service consumption [23], and distinguished professional backgrounds [24] promoted corporate tax avoidance. This disagreement in scholars' views prompted the writing of this paper.

The authors believe that all people are profitable. When managerial power reaches a certain level, executives are likely to choose personal interests over corporate values. At that time, rent-seeking behavior is very likely, and executives are likely to abandon corporate tax avoidance in order to maximize personal interests. Management controls corporate power and sets the company's tax policy. If management wants to create a better corporate governance environment and a more reasonable shareholder structure, senior management must inhibit corporate tax avoidance behavior to achieve their goals. The realization of those goals grants the management more shareholder trust, thus

granting them more managerial power and creating a virtuous cycle. Based on this, this paper proposes Hypothesis 1:

**Hypothesis 1.** *Managerial power inhibits corporate tax avoidance.*

Managerial power stems in part from their equity holdings, and more managerial power reflects a more concentrated equity system. A company with concentrated equity tends to be risk-averse and curbs its tax avoidance to reduce financial risk; this phenomenon is especially pronounced in family businesses. In addition, the public and shareholders often become very concerned when managerial power is high. The resulting supervision pressure and tax avoidance costs reduce the tendency of companies to avoid taxation. Based on this, we propose Hypothesis 2:

**Hypothesis 2.** *Firms with greater managerial power have fewer incentives to avoid tax.*

## 3. Data and Methodology

### 3.1. Sample

We collected all our financial and managerial data from the Chinese Stock Market and Accounting Research database (CSMAR). We selected Chinese firms that traded A-shares between 2000 and 2016. We excluded firms with special treatment, like ST, *ST, PT, *PT, along with data from their first year. We also removed firms in the financial industry, because financial statements in the financial industry differ greatly from other industries. Next, we excluded firms with a leverage ratio over one and negative operating revenue. Lastly, we excluded samples with missing variables, leaving a sample of 2307 firms. All the variables in our paper were Winsorized at the 1% level every year.

### 3.2. Variable Definition

#### 3.2.1. Measurements of Tax Avoidance

Most of the previous literature employed effective tax rate (ETR) and book-to-tax difference (BTD) to measure corporate tax avoidance [17,36].

We calculated ETR as (Total Income Tax Expense–Deferred Tax Expense)/Pre-Tax Book Income. Since ETR cannot distinguish between aggressive tax reporting and tax avoidance, we followed Tang et al. [37] and modified ETR to create marginal effective tax rate (METR) by dividing ETR by the actual tax rate (ATR). That is to say, METR is (Total Income Tax Expense–Deferred tax expense)/(ATR $\times$ Pre-Tax Book Income).

We utilized METR as our main independent variable, although we employed ETR as a robustness check. Since earnings-management may affect ETR, we changed its denominator from Pre-Tax Book Income to Operating Cash Flow.

An alternative measure for tax avoidance is Book-Tax Difference (BTD). However, BTD could also be affected by accrual manipulation, so we followed Kim and Zhang [36] and utilized a compound factor that combined BTD, Residual Book-Tax Difference (DDBTD), and Discretionary Permanent Book-Tax Difference (DTAX) together. DDBTD was proposed by Desai and Dharmapala [11] to exclude the effect of earnings manipulation and the firms' own characteristics. We employed the regression residuals ($\varepsilon_{i,t}$) from Equation (1):

$$BTD_{i,t+1} = \alpha + \beta_0 BTD_{i,t} + \beta_1 Total\ Accruals_{i,t} + \mu_i + \varepsilon_{i,t} \tag{1}$$

Equation (1) estimates total accruals using the modified Jones model, and $\mu_i$ represents the firm-level fixed effect.

Carter et al. [38] proposed DTAX, which excluded the impact of intangible assets (INTANG), equity investment attributed to minority shareholders (MI), equity investment (UNCON), income tax (CSTE), and the BTD of the prior year. DTAX is the regression residuals ($\varepsilon_{i,t}$) in Equation (2):

$$BTD_{i,t+1} = \alpha + \beta_0 BTD_{i,t} + \beta_1 INTANG_{i,t} + \beta_2 UNCON_{i,t} + \beta_3 MI_{i,t} + \beta_4 CSTE_{i,t} + \beta_5 BTD_{i,t-1} + \varepsilon_{i,t} \quad (2)$$

### 3.2.2. Measurements of Managerial Power

We utilized the six variables in Table 1 to measure managerial power, as each variable correlated positively with managerial power. Thus, we employed PCA and entropy weight methodology to combine the six variables together.

**Table 1.** Variable definition for managerial power.

| Variables | Definition |
| --- | --- |
| CEO-chairman duality | Dummy variable, one means that CEO is also the chairman of the board |
| Managerial shareholding | The total shareholding of top management team |
| TMT scale | The number of top executives |
| Equity dispersion | Dummy variable, one denotes that top 2–10 shareholders' shareholding is larger than the shareholding of largest shareholder |
| CEO connection | The dummy variable, one denotes CEO is the director of other firms |
| CEO's political connection | Dummy variable, one denotes that CEO has the political background |

When studying multiple variables, there is often a correlation between the two variables, creating an overlapping reflection in the results of the study. Therefore, PCA implements dimensionality reduction of the variables by recombining the original variables into a new set of mutually independent composite variables, while reflecting as much of the original information as possible. Thus, in this paper, we used Equation (3) to standardize the variables:

$$X'_{ij} = \frac{(X_{ij} - min(X_{1j}, X_{2j}, \cdots, X_{nj}))}{max(X_{1j}, X_{2j}, \cdots, X_{nj}) - min(X_{1j}, X_{2j}, \cdots, X_{nj})} + 1 \qquad (i = 1, 2, \cdots, n; j = 1, 2, \cdots, m) \quad (3)$$

We added the main components of the PCA to the entropy method, and then integrated them into a measure of managerial power. The entropy method is an objective weighting method that determines the index weight according to the amount of information provided by each index's observations. In information theory, entropy is a measure of uncertainty. The more information there is, the smaller the entropy and uncertainty. According to the characteristics of entropy, we can calculate entropy to judge the discrete degree of an index. The greater the index's discrete degree, the greater the index's impact on the comprehensive indicator. We proposed that there are $m$ programs, $n$ indexes, and formed an original data matrix $X_{i,j(m \times n)}$. For index $X_j$, the greater its gap with the index value $X_{i,j}$, the more important of a role index $X_j$ played in the comprehensive evaluation. If all the values of an index were equal, the index was not important. Therefore, according to each index's degree of variation, the weight of each index can be calculated through information entropy.

After standardizing the variables, we calculated the weight of each observation $X_{i,j}$ in the sum of variable $X_j$ using Equation (4):

$$P_{ij} = \frac{X_{ij}}{\sum\limits_{i=1}^{n} X_{ij}} \qquad (j = 1, 2, \cdots, m) \qquad (4)$$

After that, we used the Equation (5) to calculate the entropy value for each variable $X_j$:

$$e_j = -k \cdot \sum_{i=1}^{n} P_{ij} \log(P_{i,j}) \qquad (k > 0, e_j \geq 0) \qquad (5)$$

Since constant $k$ is related to the number of samples $m$, we let $k = 1/\ln m$, so that $0 \leq e \leq 1$.

We employed the entropy of variable $X_j$ to calculate the difference coefficient of $X_j \cdot g_j = 1 - e_j$ where $g_j$ denoted the importance of variable $X_j$ in the final score. Then we calculated the weight of each variable by using Equation (6):

$$W_j = \frac{g_j}{\sum\limits_{j=1}^{m} g_j} \qquad (j = 1, 2, \cdots, m) \tag{6}$$

Finally, we calculated managerial power by weighting each variable's score. The scores were between zero and one, with one representing the highest level of managerial power.

### 3.2.3. Measurements of Control Variables

Table 2 displays the definitions for the variables in Equation (7).

**Table 2.** Variable definition for Equation (7).

| Variables | Definition |
| --- | --- |
| METR | The marginal effective tax and is calculated as (taxable income—deferred tax)/ (actual tax rate (ATR) × pre-tax book income) |
| METR2 | The marginal effective tax and is calculated as (–1) × (taxable income—deferred tax)/ (actual tax rate (ATR) × operating cash flow) |
| ETR | The effective tax rate and is calculated by the equation: (Income tax expense—deferred tax)/pre-tax book income |
| BTD | The total book-tax difference and is calculated as difference between net income and taxable income, scaled by the total assets in last year |
| BTDFACTOR | The compounded factor combined by the BTD, DDBTD [11] as well as DTAX [39]. We use the PCA to get the principal factor and define this factor as the BTDFACTOR [36]. |
| Power | Measurement for managerial power and we employ the PCA to combine six factors, including CEO-chairman duality, Managerial shareholding, TMT scale, Equity dispersion, CEO connection, CEO's political connection. We employ the principal factor to measure the CEO power |
| Power2 | Alternative measure for managerial power and we employ the entropy weight methodology to combine these six factors. |
| Size | The natural logarithm of total assets of the firms |
| MTB | The ratio of market value to book value |
| Lev | The financial leverage, the ratio of total debts to total income |
| Cash Holding | The net increase of cash and cash equivalents scaled by the total assets of last year |
| ROA | The net income to the total assets |
| Equity Income | The income from the firms' investment |
| PPE | The net fixed assets scaled by total assets of last year |
| Intangible | The ratio of intangible assets to the total assets |
| Foreign | Whether firm have operating revenue from other countries or regions |
| Industry | Industry dummy variables and denote the industry fixed effect. We employ the first Two-digit of 2012-modified industry classification of China Securities Regulatory Commission. |
| Year | Year dummy variables and represent the year fixed effect |

Combining the variables introduced before, we constructed the regression model in Equation (7):

$$
\begin{aligned}
Tax\ avoidance_{i,t+1} =\ & \alpha + \beta_0 Tax\ avoidance_{i,t} + \beta_1 Size_{i,t} + \beta_2 MTB_{i,t} + \beta_3 Lev_{i,t} + \beta_4 Cash\ Holding_{i,t} \\
& + \beta_5 ROA_{i,t} + \beta_6 Equity\ Income_{i,t} + \beta_7 Intangible_{i,t} + \beta_8 Foreign_{i,t} + \gamma Power_{i,t} \\
& + \sum Industry + \sum Year
\end{aligned} \tag{7}
$$

## 4. Empirical Results

### 4.1. Description Statistic

We presented the description statistic for each variable in our regression analysis in Table 3. In this paper, we obtained 13,671 samples from the preliminary processing of the original data, and then performed descriptive statistics on each variable. As shown in Table 3, the average METR of listed companies in China was −6.072; this means that the actual tax rate was lower than the nominal tax rate.

Therefore, listed companies in China practiced tax avoidance behavior. The minimum value of METR was −263.987, and the maximum was 67.179, which is a huge difference indicating that there is a large gap in tax planning among Chinese companies. The average power value was −0.045, the minimum value was −1.557, and the maximum was 2.97. There was also a big difference in the concentration of managerial power among the listed companies. We can see from Table 3 that the profitability of listed companies in China was relatively good. While the minimum value of return on assets (ROA) was negative, its P25 was 0.011, which is much larger than zero, indicating that more than 75% of listed companies were profitable.

**Table 3.** Descriptive statistics.

| Variables | N | Mean | SD | Min | P25 | P50 | P75 | Max |
|---|---|---|---|---|---|---|---|---|
| METR | 13,671 | −6.072 | 11.774 | −263.987 | −8.118 | −4.931 | −2.472 | 67.179 |
| METR2 | 13,671 | −4.068 | 21.804 | −185.064 | −6.619 | −2.528 | −0.085 | 139.498 |
| ETR | 13,671 | 1.088 | 2.123 | −16.971 | 0.567 | 1.008 | 1.396 | 48.444 |
| BTD | 13,671 | −0.007 | 0.061 | −0.358 | −0.024 | −0.005 | 0.011 | 0.749 |
| BTDFACTOR | 13,671 | −0.013 | 0.532 | −3.4 | −0.164 | 0.008 | 0.148 | 5.36 |
| Power | 13,671 | −0.045 | 0.78 | −1.157 | −0.638 | −0.248 | 0.36 | 2.97 |
| Power2 | 13,671 | 0.355 | 0.169 | 0 | 0.264 | 0.337 | 0.449 | 0.947 |
| Size | 13,671 | 22.137 | 1.215 | 19.227 | 21.283 | 21.978 | 22.828 | 26.101 |
| MTB | 13,671 | 1.88 | 1.779 | 0.143 | 0.749 | 1.35 | 2.376 | 18.68 |
| Lev | 13,671 | 0.48 | 0.194 | 0.046 | 0.336 | 0.49 | 0.63 | 0.896 |
| Cash Holding | 13,671 | 0.023 | 0.107 | −0.28 | −0.025 | 0.008 | 0.051 | 0.868 |
| ROA | 13,671 | 0.034 | 0.052 | −0.334 | 0.011 | 0.031 | 0.057 | 0.22 |
| Equity Income | 13,671 | 0.01 | 0.022 | −0.064 | 0 | 0.002 | 0.01 | 0.182 |
| PPE | 13,671 | 0.291 | 0.206 | 0.002 | 0.135 | 0.252 | 0.411 | 1.124 |
| Intangible | 13,671 | 0.054 | 0.069 | 0 | 0.015 | 0.036 | 0.066 | 0.598 |
| Foreign | 13,671 | 0.494 | 0.5 | 0 | 0 | 0 | 1 | 1 |

*4.2. Correlation Analysis*

We carried out a Pearson correlation analysis and showed the results in Table 4. Both Power and Power2 had a negative relationship with METR, BTD, and BTDFACTOR, and a positive relationship with ETR. This result was consistent with the expectations of Hypothesis 1, and established that firms with high managerial power had fewer incentives to avoid tax. The correlation coefficient between ROA and METR was −0.046, indicating that a company's ROA had a statistically significant negative correlation with METR, and a significant, positive correlation with the degree of corporate tax avoidance.

**Table 4.** Pearson correlation analysis (to be continued).

| Variables | METR1 | METR2 | ETR1 | BTD | BTDFACTOR | Power | Power2 |
|---|---|---|---|---|---|---|---|
| METR1 | 1 | | | | | | |
| METR2 | 0.126 *** | 1 | | | | | |
| ETR | −0.971 *** | −0.115 *** | 1 | | | | |
| BTD | 0.356 *** | 0.104 *** | −0.369 *** | 1 | | | |
| BTDFACTOR | 0.354 *** | 0.100 *** | −0.366 *** | 0.987 *** | 1 | | |
| Power | −0.034 *** | −0.030 *** | 0.017 ** | −0.005 | 0 | 1 | |
| Power2 | −0.041 *** | −0.023 *** | 0.031 *** | −0.021 ** | −0.016 * | 0.713 *** | 1 |
| Size | −0.005 | 0 | 0.024 *** | 0.021 ** | 0.014 * | 0.061 *** | 0.251 *** |
| MTB | 0.008 | −0.030 *** | −0.030 *** | 0.058 *** | 0.083 *** | 0.126 *** | −0.043 *** |
| Lev | 0.002 | 0.044 *** | 0.025 *** | −0.143 *** | −0.156 *** | −0.084 *** | 0.076 *** |
| Cash Holding | 0.014 | −0.016* | −0.014 | 0.057 *** | 0.073 *** | 0.044 *** | 0.020 ** |
| ROA | −0.046 *** | −0.091 *** | 0.042 *** | 0.360 *** | 0.397 *** | 0.077 *** | 0.055 *** |
| Equity Income | 0.053 *** | −0.022 ** | −0.051 *** | 0.227 *** | 0.221 *** | −0.029 *** | −0.044 *** |
| PPE | 0.054 *** | 0.009 | −0.051 *** | 0.031 *** | 0.033 *** | −0.052 *** | 0.012 |
| Intangible | 0.032 *** | 0.004 | −0.033 *** | 0.022 ** | −0.047 *** | 0.056 *** | 0.049 *** |
| Foreign | −0.043 *** | −0.027 *** | 0.012 | −0.006 | −0.004 | 0.088 *** | 0.080 *** |

Note: *** $p < 0.01$, ** $p < 0.05$, * $p < 0.1$.

From Table 5, we can see that Lev had a statistically significant positive correlation with firm size and a significant, negative correlation with MTB, indicating that the larger the firm size, the larger the bond ratio, but the smaller the MTB. At the same time, equity income was positively correlated with MTB and cash holdings, and negatively correlated with Lev, indicating that the larger the MTB and cash holding companies, the greater the income that shareholders received. In addition, we can see from Table 5 that a company's net fixed assets were positively correlated with the company's cash holdings, but negatively correlated with ROA, while intensive asset intensity had a significant, positive correlation with cash holdings and ROA.

**Table 5.** Pearson correlation analysis (continued from Table 4).

| Variables | Size | MTB | Lev | Cash Holding | ROA | Equity Income | PPE | Intangible |
|---|---|---|---|---|---|---|---|---|
| Size | 1 | | | | | | | |
| MTB | −0.400 *** | 1 | | | | | | |
| Lev | 0.418 *** | −0.495 *** | 1 | | | | | |
| Cash Holding | 0.063 *** | 0.093 *** | −0.026 *** | 1 | | | | |
| ROA | 0.063 *** | 0.267 *** | −0.356 *** | 0.146 *** | 1 | | | |
| Equity Income | 0.013 | 0.097 *** | −0.129 *** | 0.095 *** | 0.231 *** | 1 | | |
| PPE | 0.097 *** | −0.146 *** | 0.055 *** | 0.024 *** | −0.024 *** | −0.147 *** | 1 | |
| Intangible | 0.021 ** | 0.041 *** | −0.047 *** | 0.086 *** | 0.028 *** | −0.014 | 0.107 *** | 1 |
| Foreign | 0.050 *** | 0.015 * | −0.030 *** | −0.004 | −0.007 | −0.039 *** | −0.006 | −0.020 ** |

Note: *** $p < 0.01$, ** $p < 0.05$, * $p < 0.1$.

### 4.3. Baseline Result

Table 6 displays the results of our OLS regression. We controlled for the industry and year fixed effects in each model. We included Power in Model 1 and included other control variables in Model 2. Both models passed the F test at the 1% level, indicating they had statistical significance. Power had a negative coefficient that was significant at the 5% level. This result indicated that managerial power reduced corporate tax avoidance, in line with Hypothesis 2. A possible reason is that tax avoidance benefited shareholders, but it was harmful for executives.

**Table 6.** Baseline results.

| | (1) | (2) |
|---|---|---|
| | **Model1** | **Model2** |
| Variables | METR | METR |
| Size | | 0.222 |
| | | (1.565) |
| MTB | | 0.263 *** |
| | | (4.219) |
| Lev | | −1.630 ** |
| | | (−2.083) |
| Cash Holding | | 0.823 |
| | | (1.017) |
| ROA | | −18.202 *** |
| | | (−9.076) |
| Equity Income | | 35.682 *** |
| | | (7.865) |
| PPE | | 3.710 *** |
| | | (6.145) |
| Intangible | | 5.403 *** |
| | | (3.236) |
| Foreign | | −0.381 |
| | | (−1.527) |

**Table 6.** *Cont.*

|  | (1) | (2) |
|---|---|---|
| Power | −0.369 *** | −0.322 ** |
|  | (−2.952) | (−2.536) |
| Observations | 13,671 | 13,671 |
| R-squared | 0.020 | 0.031 |
| Year FE | Yes | Yes |
| Industry FE | Yes | Yes |
| F | 8.716 | 20.80 |

The t-statistics in parentheses are adjusted by clustering at the firm level; [2] Note: *** $p < 0.01$, ** $p < 0.05$, * $p < 0.1$.

## 4.4. Robustness Tests

We conducted a series of robustness checks. First, we employed an alternative measure for managerial power by utilizing entropy weight methodology. Next, we harnessed an alternative measure for tax avoidance. Finally, we utilized other methodology such as quantile regression and propensity score matching (PSM).

### 4.4.1. Alternative Measures of Managerial Power

First, we calculated managerial power using entropy weight methodology, listing the results on Table 7. Power had a negative coefficient, significant at the 1% level. The results were consistent with the baseline model, illustrating the robustness of our results.

**Table 7.** Robust check with alternative measure for managerial power.

|  | (1) | (2) |
|---|---|---|
|  | **Model1** | **Model2** |
| **Variables** | **METR** | **METR** |
| Size |  | 0.278 * |
|  |  | (1.951) |
| MTB |  | 0.258 *** |
|  |  | (4.152) |
| Lev |  | −1.565 ** |
|  |  | (−2.012) |
| Cash Holding |  | 0.764 |
|  |  | (0.945) |
| ROA |  | −18.146 *** |
|  |  | (−9.056) |
| Equity Income |  | 35.557 *** |
|  |  | (7.803) |
| PPE |  | 3.727 *** |
|  |  | (6.193) |
| Intangible |  | 5.407 *** |
|  |  | (3.242) |
| Foreign |  | −0.380 |
|  |  | (−1.529) |
| Power2 | −2.188 *** | −1.976 *** |
|  | (−2.955) | (−2.715) |
| Observations | 13,671 | 13,671 |
| R-squared | 0.020 | 0.031 |
| Year FE | Yes | Yes |
| Industry FE | Yes | Yes |
| F | 8.733 | 20.70 |

Note: The t-statistics in parentheses are adjusted by clustering at the firm level; *** $p < 0.01$, ** $p < 0.05$, * $p < 0.1$.

### 4.4.2. Alternative Measures of Tax Avoidance

We employed four other tax avoidance measures as proxies for tax avoidance, the results were showed in Table 8. For these measures, a higher ETR value means less tax avoidance, so that Power has a positive and significant coefficient in Model 2. For the remaining measures, higher values denote stronger tax avoidance, and the coefficients of Power were negative in those models. In a word, these results were consistent with the previous results, demonstrating their robustness.

**Table 8.** Robust check with alternative measure for tax avoidance.

| Variables | (1) Model1 METR2 | (2) Model2 ETR | (3) Model3 BTD | (4) Model4 BTDFACTOR |
|---|---|---|---|---|
| Size | 0.083 | −0.022 | −0.001 | −0.015 * |
| | (0.377) | (−0.886) | (−1.390) | (−1.824) |
| MTB | 0.182 | −0.039 *** | −0.004 *** | −0.025 *** |
| | (1.280) | (−3.554) | (−6.267) | (−5.124) |
| Lev | 0.003 | 0.429 *** | −0.014 *** | −0.099 *** |
| | (0.002) | (3.099) | (−3.304) | (−2.770) |
| Cash Holding | −1.104 | −0.137 | −0.009 | 0.004 |
| | (−0.606) | (−0.924) | (−1.289) | (0.072) |
| ROA | −37.274 *** | 3.312 *** | 0.409 *** | 3.940 *** |
| | (−9.604) | (9.201) | (23.794) | (28.392) |
| Equity Income | −2.649 | −6.492 *** | 0.431 *** | 3.362 *** |
| | (−0.189) | (−7.892) | (7.843) | (7.384) |
| PPE | 2.424 ** | −0.627 *** | 0.020 *** | 0.214 *** |
| | (2.395) | (−5.874) | (4.723) | (5.929) |
| Intangible | 4.093 | −1.079 *** | 0.027 ** | −0.334 *** |
| | (1.518) | (−3.593) | (2.149) | (−3.131) |
| Foreign | −0.319 | 0.023 | 0.002 | 0.017 |
| | (−0.670) | (0.553) | (1.485) | (1.540) |
| Power | −0.489 * | 0.038 * | −0.002 ** | −0.012 ** |
| | (−1.842) | (1.806) | (−2.530) | (−2.170) |
| Observations | 13,671 | 13,671 | 13,671 | 13,671 |
| R-squared | 0.014 | 0.028 | 0.186 | 0.211 |
| Year FE | Yes | Yes | Yes | Yes |
| Industry FE | Yes | Yes | Yes | Yes |
| F | 15.25 | 20.14 | 86.27 | 115.5 |

Note: The t-statistics in parentheses are adjusted by clustering at the firm level; *** $p < 0.01$, ** $p < 0.05$, * $p < 0.1$.

### 4.4.3. Different Quantile Regression

Since OLS regression assumes that the explanatory variables only affect the mean position of the conditional distribution of the interpreted variable, OLS will not be robust when there is a spike or thick tail distribution in the data, or if the data has significant heteroscedasticity [40]. In that case, quantile regression can more accurately describe the influence of the explanatory variable over its range of variation and the shape of the conditional distribution, and quantile regression does not require strong assumptions for the error term. Therefore, for a non-normal distribution, quantile regression coefficient estimation results are more robust than OLS regression.

In the quantile regression results showed in Table 9, Power had a negative and significant coefficient, indicating that managerial power reduced firm tax avoidance. This is consistent with Dowling [15], who thought that, even if the level of corporate governance was low, it could reduce firm tax avoidance.

**Table 9.** Robust check: quantile regression.

| | Q1/4 | Q1/2 | Q3/4 |
|---|---|---|---|
| **Variables** | **Model 1**<br>**METR** | **Model 2**<br>**METR** | **Model 3**<br>**METR** |
| Size | 0.207 * | 0.209 *** | 0.108 * |
| | (1.684) | (3.117) | (1.811) |
| MTB | 0.198 *** | 0.226 *** | 0.189 *** |
| | (3.089) | (7.201) | (5.518) |
| Lev | −1.939 *** | −0.816 ** | −0.457 |
| | (−2.717) | (−2.047) | (−1.349) |
| Cash Holding | 0.093 | −0.062 | −0.195 |
| | (0.163) | (−0.210) | (−0.602) |
| ROA | 3.012 | −22.362 *** | −27.207 *** |
| | (1.052) | (−19.644) | (−24.733) |
| Equity Income | 23.907 *** | 35.812*** | 35.927 *** |
| | (5.906) | (16.166) | (18.277) |
| PPE | 2.161 *** | 1.801 *** | 1.620 *** |
| | (4.148) | (6.020) | (5.324) |
| Intangible | 0.700 | 1.103* | 1.409 ** |
| | (0.620) | (1.789) | (2.388) |
| Foreign | −0.422 ** | −0.376 *** | −0.157 |
| | (−2.055) | (−2.741) | (−1.460) |
| Power | −0.302 *** | −0.296 *** | −0.303 *** |
| | (−3.025) | (−5.105) | (−4.959) |
| Observations | 13,671 | 13,671 | 13,671 |
| R-squared | 0.014 | 0.018 | 0.012 |
| Year FE | Yes | Yes | Yes |
| Industry FE | Yes | Yes | Yes |

Note: The t-statistics in parentheses are adjusted by clustering at the firm level; *** $p < 0.01$, ** $p < 0.05$, * $p < 0.1$.

### 4.4.4. Propensity Score Matching

PSM has two important assumptions that OLS does not consider. One is the common support hypothesis, that the two sets of samples should have similar characteristics as a whole. The other is the balancing assumption, that the two sets of samples should be similar in each dimension. Therefore, we employed a non-parameter methodology, PSM, as an alternative test for the impact of managerial power [41]. We divided the sample into two parts according to the average managerial power in each year and industry. In the results for the unmatched sample which were showed in Table 10, firms with high-power executives had less inclination towards tax avoidance. We employed logit and probit models to estimate the propensity score. We utilized matching with replacement and without replacement, as well as one-to-one and one-to-four matching. All the results showed that firms with high managerial power were less likely to avoid taxes, in line with Hypothesis 2. This result is consistent with Slemrod [18], who pointed out that corporate tax avoidance could be a self-interested behavior tool for managers, who used their power to decrease tax avoidance behavior in order to maximize their own self-interest.

**Table 10.** Robust check: propensity score matching.

| Type | | Replacement | Pair | Low Power | High Power | Diff | Std. Err | T Statistic |
|---|---|---|---|---|---|---|---|---|
| Unmatched | | | | −6.3584 | −5.8665 | −0.4919 | 0.2042 | −2.41 ** |
| Matched | Logit | Yes | 1:1 | −6.3584 | −5.7902 | −0.5682 | 0.2601 | −2.18 ** |
| | | Yes | 1:4 | −6.3584 | −5.8899 | −0.4686 | 0.2254 | −2.08 ** |
| | | No | 1:1 | −6.3584 | −5.8751 | −0.4833 | 0.2108 | −2.29 ** |
| | Probit | Yes | 1:1 | −6.3584 | −5.8747 | −0.4837 | 0.2677 | −1.81 * |
| | | Yes | 1:4 | −6.3584 | −5.9683 | −0.3901 | 0.2231 | −1.75 * |
| | | No | 1:1 | −6.3584 | −5.9679 | −0.3905 | 0.2120 | −1.84 * |

Note: *** $p < 0.01$, ** $p < 0.05$, * $p < 0.1$.

### 4.4.5. Controlling for Endogeneity

Our sample may have endogeneity problems. For example, firms that do more to avoid taxes may restrict managerial power so that shareholders can gain more tax benefits. To address this concern, we conducted an instrumental variable regression. We followed Lin Song [42] and used the average managerial power in each year and industry as the instrumental variable. It is unlikely that industrial managerial power levels would affect firm tax avoidance through other channels, so we conducted a 2SLS instrumental regression and presented the results on Table 11. 2SLS is a single equation estimation method that is suitable for both structural equations and for over-identification [43]. We can see that the F statistic is larger than 10 in the first-stage regression, indicating that there is no weak instrumental problem. In the second-stage results, Power still had negative and significant coefficient, illustrating the robustness of our results.

**Table 11.** Robust check: endogeneity tests.

|  | First Stage | Second Stage | First Stage | Second Stage |
|---|---|---|---|---|
| **Variables** | **Model 1** **Power** | **Model 2** **METR** | **Model 3** **Power** | **Model 4** **METR** |
| Power IV | 1.000 *** | | 0.950 *** | |
|  | (12.885) | | (12.083) | |
| Predicted Power | | −2.534 *** | | −2.726 *** |
|  | | (−2.635) | | (−2.672) |
| Size | | | 0.061 *** | 0.365 ** |
|  | | | (3.572) | (2.375) |
| MTB | | | 0.036 *** | 0.358 *** |
|  | | | (4.027) | (4.710) |
| Lev | | | −0.115 | −1.940 ** |
|  | | | (−1.415) | (−2.384) |
| Cash Holding | | | 0.144 ** | 1.203 |
|  | | | (2.293) | (1.396) |
| ROA | | | 0.529 ** | −16.880 *** |
|  | | | (2.239) | (−7.738) |
| Equity Income | | | −1.838 *** | 30.924 *** |
|  | | | (−4.018) | (6.086) |
| PPE | | | −0.112 | 3.480 *** |
|  | | | (−1.527) | (5.487) |
| Intangible | | | 0.512 *** | 6.635 *** |
|  | | | (2.783) | (3.731) |
| Foreign | | | 0.049 | −0.249 |
|  | | | (1.629) | (−0.910) |
| Observations | 13,671 | 13,671 | 13,671 | 13,671 |
| R-squared | 0.089 | 0.001 | 0.104 | 0.008 |
| Year FE | Yes | Yes | Yes | Yes |
| Industry FE | Yes | Yes | Yes | Yes |
| F | 166.0 | 6.942 | 22.14 | 19.37 |

Note: The t-statistics in parentheses are adjusted by clustering at the firm level; *** $p < 0.01$, ** $p < 0.05$, * $p < 0.1$.

### 4.4.6. Simultaneous Regression Equation

A simultaneous regression equation is a regression model that incorporates all the explanatory variables into the equation to estimate their influence. Therefore, the simultaneous regression equation provides the overall interpretation of all explanatory variables. We employed a simultaneous equation model to test our causal relationship and showed the results in Table 12. In the results, Power reduced METR significantly, while METR had no significant effect on Power, indicating that Power had a negative casual effect on tax avoidance, consistent with our baseline result.

**Table 12.** Robust check: simultaneous regression equation.

| | (1) | (2) |
|---|---|---|
| **Variables** | **METR** | **Power** |
| Power | −2.726 ** | |
| | (−2.407) | |
| Power IV | | 1.411 ** |
| | | (2.291) |
| METR | | 0.178 |
| | | (0.788) |
| Size | 0.365 *** | 0.025 |
| | (2.711) | (0.499) |
| MTB | 0.358 *** | −0.010 |
| | (3.891) | (−0.166) |
| Lev | −1.940 *** | 0.175 |
| | (−2.677) | (0.449) |
| Cash Holding | 1.203 | |
| | (1.215) | |
| ROA | −16.880 *** | 3.791 |
| | (−7.053) | (0.921) |
| Equity Income | 30.924 *** | −8.236 |
| | (5.704) | (−1.000) |
| PPE | 3.480 *** | −0.785 |
| | (5.707) | (−0.902) |
| Intangible | 6.635 *** | −0.421 |
| | (4.017) | (−0.338) |
| Foreign | −0.249 | 0.117 |
| | (−1.032) | (1.210) |
| Observations | 13,671 | 13,671 |
| R-squared | 0.008 | −6.996 |
| Year FE | Yes | Yes |
| Industry FE | Yes | Yes |

Note: The t-statistics in parentheses are adjusted by clustering at the firm level; *** $p < 0.01$, ** $p < 0.05$, * $p < 0.1$.

### 4.4.7. Dynamic Analysis

We reconstructed the BTDFACTOR and conducted the robust check. The results were shown as following Table 13. The coefficient of Power was still negative and significant, illustrating the robustness of our result.

**Table 13.** Dynamic analysis for Equation (2).

| | (1) |
|---|---|
| | **Model1** |
| **Variables** | **BTDFACTOR** |
| Size | −0.006 |
| | (−0.872) |
| MTB | 0.006 |
| | (1.234) |
| Lev | −0.028 |
| | (−0.908) |
| Cash Holding | −0.001 |
| | (−0.022) |
| ROA | 1.068 *** |
| | (9.071) |
| Equity Income | 1.478 *** |
| | (4.058) |
| PPE | 0.092 *** |
| | (3.206) |

**Table 13.** *Cont*.

|  | **(1)** |
| --- | --- |
| Intangible | −0.235 *** |
|  | (−3.493) |
| Foreign | −0.002 |
|  | (−0.177) |
| Power | −0.009 * |
|  | (−1.678) |
| Observations | 11,540 |
| R-squared | 0.061 |
| Year FE | Yes |
| Industry FE | Yes |
| F | 16.41 |

Note: The t-statistics in parentheses are adjusted by clustering at the firm level; *** $p < 0.01$, ** $p < 0.05$, * $p < 0.1$.

For Equation (7), we conducted the robust check and replaced the dependent variable in year $t$ with the tax avoidance in year $t+1$. Besides, we included the tax avoidance in year $t$ as the control variable. The result was shown in Table 14. Power still had a negative and significant coefficient regression. The result was still consistent with prior result, indicating the robustness of our result.

**Table 14.** Dynamic analysis for Equation (7).

|  | **(1)** | **(2)** | **(3)** | **(4)** |
| --- | --- | --- | --- | --- |
| **Variables** | **Model1** $METR1_{t+1}$ | **Model2** $METR1_{t+1}$ | **Model3** $METR1_{t+1}$ | **Model4** $METR1_{t+1}$ |
| $METR1_t$ | −0.031 | −0.035 | −0.031 | −0.035 |
|  | (−0.726) | (−0.836) | (−0.726) | (−0.836) |
| Power | −0.432 *** | −0.434 *** | −0.432 *** | −0.434 *** |
|  | (−3.168) | (−3.141) | (−3.168) | (−3.141) |
| Size |  | −0.216 |  | −0.216 |
|  |  | (−1.340) |  | (−1.340) |
| MTB |  | 0.264 *** |  | 0.264 *** |
|  |  | (2.740) |  | (2.740) |
| Lev |  | 1.132 |  | 1.132 |
|  |  | (1.159) |  | (1.159) |
| Cash Holding |  | 1.450 |  | 1.450 |
|  |  | (1.299) |  | (1.299) |
| ROA |  | −0.826 |  | −0.826 |
|  |  | (−0.293) |  | (−0.293) |
| Equity Income |  | 17.191 ** |  | 17.191 ** |
|  |  | (2.302) |  | (2.302) |
| PPE |  | 2.925 *** |  | 2.925 *** |
|  |  | (4.582) |  | (4.582) |
| Intangible |  | 1.687 |  | 1.687 |
|  |  | (1.027) |  | (1.027) |
| Foreign |  | −0.501 * |  | −0.501 * |
|  |  | (−1.738) |  | (−1.738) |
| Observations | 10,655 | 10,655 | 10,655 | 10,655 |
| R-squared | 0.024 | 0.027 | 0.024 | 0.027 |
| Year FE | Yes | Yes | Yes | Yes |
| Industry FE | Yes | Yes | Yes | Yes |
| F | 5.362 | 4.636 | 5.362 | 4.636 |

Note: The t-statistics in parentheses are adjusted by clustering at the firm level; *** $p < 0.01$, ** $p < 0.05$, * $p < 0.1$.

## 5. Conclusions and Discussion

This paper studied managerial power and corporate tax avoidance. In order to examine this relationship in detail, we used PCA to construct managerial power, and OLS regression to

explore the relationship. Then we used quantile regression, PSM, instrumental variable regression, and simultaneous regression to check the model's robustness. In the end, we came to the following two conclusions:

(1) Managerial power reduces corporate tax avoidance, and firms with greater managerial power have lesser incentive to avoid taxes. From our models, the correlation coefficient between managerial power and corporate tax avoidance was −0.369 before adding any control variables, and −0.322 after adding control variables, which indicated a significant and inhibitory effect of managerial power on corporate tax avoidance. We utilized PSM to divide the sample into two parts according to average managerial power. Logit and probit regressions showed that managerial power reduced METR significantly, while METR had no significant effect on power, indicating that managerial power had a negative casual effect on tax avoidance. In addition, the correlation coefficient between ROA and METR was −0.046, indicating that the ROA of a company was significant and negatively correlated with METR, so a company's ROA was significant and positively correlated with its degree of corporate tax avoidance. We can assume that management gave up the maximization of corporate value for the sake of self-interest, thereby suppressing the tax avoidance behavior of the company through self-interested behaviors such as rent seeking [11]; that is, the reduction of corporate ROA and the promotion of personal interests. The authors believe that management that had more power was inclined towards better corporate governance [16] and shareholding structure reform [17], and that it is necessary to curb corporate tax avoidance behavior to achieve these goals. The authors' future research will be on the relationship between equity governance and corporate tax avoidance.

(2) Companies need to improve managerial power through appropriate means to achieve full taxation, but also strengthen supervision and restraint on the executive's self-interested behaviors. From the data argumentation, the managerial power is conducive to restraining corporate tax avoidance behaviors, so improving the managerial power can achieve full taxation, thus improving the financial system environment and the public image of companies. However, we can also see the negative correlation between ROA and METR. Therefore, behind the full taxation, the self-interest behaviors of the management may be hidden, they may fully utilize their greater power to maximize the personal interests [11]. Therefore, in order to maximize the value of the shareholders, they should increase the supervision while improving the managerial power, so as to better achieve the balance between corporate taxation and shareholder value maximization.

The research conclusions above provide some contributions to the theoretical and empirical aspects of this study. In the past, most scholars had discussed the issue of corporate tax avoidance and the relationship between executive behavior and financial quality [11–17,33] only in theory. This paper opens up new possibilities for research on enriching tax avoidance behavior and managerial power. In practice, taxation is an integral part of public policy. Tax revenues affect a country's fiscal revenues and expenditures, as well as its economic cycles and national defense security. Therefore, managing the taxation behavior of enterprises has great benefits. We can see that, in order to prompt companies to fulfill their tax obligations and reduce the practice of corporate tax avoidance, corporate shareholders need to increase senior managerial power in corporate governance. This leads to increased tax payments, thereby improving the company's image among consumers and achieving sustainable and healthy development of the company. Regulators, when conducting business guidance, should focus on those companies whose managers have too little managerial power. Regulators should promote good taxation behavior in companies by improving the power of corporate management, which reduces corporate tax avoidance behaviors.

This paper still has some limitations. First, endogenous problems with omitted variables still exist. For example, executive compensation and state ownership in the company may affect the managers' incentives to avoid taxes, but we did not include those factors in the regression model, leading to a possible endogeneity problem. Second, there is still some uncertainty when measuring tax avoidance. We still cannot completely exclude the role of management earnings incentives in tax avoidance. Third, although we utilized an instrument variable regression to tackle possible endogeneity, this

methodology is still not enough to address the problem completely. Future research should employ methodologies such as natural experiments and regression discontinuity designs to avoid any possible problems with endogeneity.

**Author Contributions:** All authors contributed significantly to the manuscript. Y.T. conceived of the concept, designed the empirical research, and co-wrote the manuscript. Y.L. took charge of designing the empirical research, collecting data and analysis, and co-wrote the manuscript. J.L. co-wrote the manuscript and contributed to the formatting of the paper. W.L. provided conceptual comments and contributed to the paper's review.

**Funding:** This research was funded by the National Natural Science Foundation of China (No. 71072066), the Ministry of Education (No. ASCZXX-01, No. 17YJC790094), Sichuan University (No. SKGT201602, No. 2018HHF-42), and the Department of Science and Technology of Sichuan Province (No. 2018JY0594).

**Acknowledgments:** Helpful comments from the reviewers are appreciated.

**Conflicts of Interest:** The authors declare that they have no conflicts of interest.

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
