# Peer review of "Does More Managerial Power Impede or Promote Corporate Tax Avoidance? Evidence from Listed Chinese Companies"

_sustainability, doi:10.3390/su11071914_

Round 1

Reviewer 1 Report

Dear Authors,

thank you for the possibility to read the paper.

I have some comments to the content and style of presentation:

1/ Please, check the English language - there are several typos/grammar errors/stylistics imperfections, thus the paper needs a thorough language check prior publication.

2/ It seems to me that to state in the introduction - line 76/77 - that the paper brings an outstanding contribution is too strong. I am aware that you as authors are persuaded about this, but to my ears it sound too strong. Consider softening the language style.

3/ Although the methodology and calculation of TA and MP are limited by the data obtained from CSMAR, I do not consider this part a weakness. Nevertheless, I have strong objections to the way have the two hypotheses were stated (as well as to the non-existent discussion to research findings). It is not clear why/how authors have put together the two hypotheses. You need to separate the both hypotheses and address their backup argumentation based on specific findings (or gaps) in the prior literature. Please, do this for both hypotheses individually.

4/ As mentioned above, there is no discussion to the research findings. The Conclusion does not help in this either. Without a comprehensive discussion to your findings, the paper remains just a statistical exercise.

5/ When writing the Discussion section, include also your answer on: What are the theoretical and practical implications? Be specific and exhaustive in this.

Kind regards,

Reviewer

Author Response

Point 1: Please, check the English language - there are several typos/grammar errors/ stylistics imperfections, thus the paper needs a thorough language check prior publication.

Response 1:

    The new version of paper not only revised the content, but also edited and polished the English style. At the same time, I also handed the paper to professional institutions for retouching, but due to the time relationship, the paper is still being edited, and the retouched paper will be submitted in the next round of revision. I am very sorry about this. Please forgive.

Point 2: It seems to me that to state in the introduction - line 76/77 - that the paper brings an outstanding contribution is too strong. I am aware that you as authors are persuaded about this, but to my ears it sound too strong. Consider softening the language style.

Response 2:

    The outstanding contributions of line 76/77 had been amended to some contributions, and the language style of other parts had been appropriately softened too.

Point 3: Although the methodology and calculation of TA and MP are limited by the data obtained from CSMAR, I do not consider this part a weakness. Nevertheless, I have strong objections to the way have the two hypotheses were stated (as well as to the non-existent discussion to research findings). It is not clear why/how authors have put together the two hypotheses. You need to separate the both hypotheses and address their backup argumentation based on specific findings (or gaps) in the prior literature. Please, do this for both hypotheses individually.

Response 3:

    The revised paper had logically combed the supporting documents of the two hypotheses, and individually expressed the background of the two hypotheses. I hope that such expressions can gain more understanding from readers.

Point 4: As mentioned above, there is no discussion to the research findings. The conclusion does not help in this either. Without a comprehensive discussion to your findings, the paper remains just a statistical exercise.

Response 4:

    The revised paper had fully discussed the statistical data and empirical results of this paper to avoid the pale conclusion.

Point 5: When writing the Discussion section, include also your answer on: What are the theoretical and practical implications? Be specific and exhaustive in this.

Response 5:

    In the 4th paragraph of the Section5, the implications of this paper are explained in detail from the theoretical and practical aspects. In terms of practical implications, new public policy perspective has been introduced for readers can feel more about the value of this paper.

Reviewer 2 Report

This is a paper of interest on a topical subject.

The data seem to be interesting, and appropriate to the problem under consideration.

The models presented in equations (1), (2) and (7) are simple, and lack dynamics (that is, there are no lagged effects).

The dependent variable in equation (2) is missing 2 subscripts, while the last 2 parameters in equation (7) seem to be presented incorrectly.

There are no diagnostic checks given in Tables 6, 7, 8, 9, 11, 12, so it is not possible to determine the robustness of the empirical results.

The final section should have greater discussion on public policy considerations.

Author Response

Point 1: The models presented in equations (1), (2) and (7) are simple, and lack dynamics (that is, there are no lagged effects).

Response 1:

     For the Equation1 and Equation2, we didn’t include the lagged term in the equation. Because previous literature, like Kim and et. al (2011), also didn’t include the lagged term in the model. It is a better way if we control the lagged term. Thus, we redesign the Equation1 and 2 as following:

    BTDi,t+1α +β0BTDi,t+β1Total Accuralsi,ti +εi,t (1)

    BTDi,t+1α +β0BTDi,t+β1INTANGi,t+β2UNCONi,t+β3MIi,t+β4CSTEi,t+β5BTDi,t-1i,t      (2)

     Thus, we reconstruct the BTDFACTOR and conduct the robust check. The results are shown as following table. The coefficient of power is still negative and significant, illustrating the robustness of our result.

(1)

Model1

VARIABLES

BTDFACTOR

Size

-0.006

(-0.872)

MTB

0.006

(1.234)

Lev

-0.028

(-0.908)

Cash Holding

-0.001

(-0.022)

ROA

1.068***

(9.071)

Equity income

1.478***

(4.058)

PPE

0.092***

(3.206)

Intangible

-0.235***

(-3.493)

Foreign

-0.002

(-0.177)

power

-0.009*

(-1.678)

Observations

11,540

R-squared

0.061

Year FE

Yes

Industry FE

Yes

F

16.41

Robust t-statistics in parentheses

*** p<0.01, ** p<0.05, * p<0.1

    For the Equation7, we conducted the robust checks and replace the dependent variable in year t with the tax avoidance in year t+1. Besides, we included the tax avoidance in year t as the control variable. The Equation7 is shown as following: 

Tax avoidancei,t+1= α +β0Tax avoidancei,t1Sizei,t+β2MTBi,t+β3Levi,t+β4Cach  Holdingi,t+β5ROAi,t   +β6Equity incomei,t+β7Intangiblei,t+β8Foreigni,tPoweri,t+∑Industry+∑Year  (7)

    And the result is shown in Table14. In the results, Power still have a negative and significant coefficient regression results is still consistent with prior result, indicating the robustness of our result.

(1)

(2)

(3)

(4)

Model1

Model2

Model3

Model4

VARIABLES

METR1t+1

METR1t+1

METR1t+1

METR1t+1

METR1t

-0.031

-0.035

-0.031

-0.035

(-0.726)

(-0.836)

(-0.726)

(-0.836)

Power

-0.432***

-0.434***

-0.432***

-0.434***

(-3.168)

(-3.141)

(-3.168)

(-3.141)

Size

-0.216

-0.216

(-1.340)

(-1.340)

MTB

0.264***

0.264***

(2.740)

(2.740)

Lev

1.132

1.132

(1.159)

(1.159)

Cash Holding

1.450

1.450

(1.299)

(1.299)

ROA

-0.826

-0.826

(-0.293)

(-0.293)

Equity Income

17.191**

17.191**

(2.302)

(2.302)

PPE

2.925***

2.925***

(4.582)

(4.582)

Intangible

1.687

1.687

(1.027)

(1.027)

Foreign

-0.501*

-0.501*

(-1.738)

(-1.738)

Observations

10,655

10,655

10,655

10,655

R-squared

0.024

0.027

0.024

0.027

Year FE

Yes

Yes

Yes

Yes

Industry FE

Yes

Yes

Yes

Yes

F

5.362

4.636

5.362

4.636

Robust t-statistics in parentheses

*** p<0.01, ** p<0.05, * p<0.1

Point 2: The dependent variable in equation (2) is missing 2 subscripts, while the last 2 parameters in equation (7) seem to be presented incorrectly.

Response 2: Equation (2) and Equation (7) had been corrected.

BTDi,t+1= α +β0BTDi,t1INTANGi,t2UNCONi,t3MIi,t4CSTEi,t5BTDi,t-1i,t      (2)

Tax avoidancei,t+1= α +β0Tax avoidancei,t1Sizei,t2MTBi,t3Levi,t4Cach  Holdingi,t5ROAi,t   6Equity incomei,t7Intangiblei,t8Foreigni,t+γPoweri,t+∑Industry+∑Year  (7)

Point 3: There are no diagnostic checks given in Tables 6, 7, 8, 9, 11, 12, so it is not possible to determine the robustness of the empirical results.

Response 3:

For the diagnostic checks, we supplement the following test.

1) heteroskedasticity problem

    For the heteroskedasticity problem, we have employed the heteroskedasticity-robust standard error to correct the possible heteroskedasticity in every equation, ensuring that our results are not affected by the heteroskedasticity problem.

2) The unusual or outlier data

    To alleviate the possible problem caused by the outlier, we have winsorized all the variables at 1% level by each year.

3) Multicollinearity problem

We calculate the variance inflation factor and found that all the VIF of equations are smaller than 5, indicating that the multicollinearity problem would not affect the results.

Point 4: The final section should have greater discussion on public policy considerations.

Response 4:

    In the 4th paragraph of the Section5, the implications of this paper are explained in detail from the theoretical and practical aspects. In terms of practical implications, new public policy perspective has been introduced for readers can feel more about the value of this paper.

Reviewer 3 Report

  I have red several times the abstract, introduction, hypothesis, results and conclusions. But still can not clearly understand the statement of the problem. There is some ambiguity of expression or logics in all these. If the higher managerial power, the lower incentive to avoid taxes. Thus, so why it should be suppressed? Do not understand. It should be presented clearly.

The introduction  and the literature review should be more consequent (it is nor so much the content issue as structure of logic it is linked).

The Hypotheses, I think, should be reformulated, using the same semantics, definitions to make it scientifically clear, non ambiguous.

The discussion at the end of the paper should be developed further to discuss the results in the light of different taxation systems, different economic levels, revealing the specificity of results.

I would suggest to improve conclusions, while expressing more clearly the final resolutions.

I'm not qualified to judge English language and style, but even for me it is clear that English language style should be improved.

There many small redaction mistakes to be corrected.

Author Response

Point 1: I have red several times the abstract, introduction, hypothesis, results and conclusions. But still can not clearly understand the statement of the problem. There is some ambiguity of expression or logics in all these. If the higher managerial power, the lower incentive to avoid taxes. Thus, so why it should be suppressed? Do not understand. It should be presented clearly.

Response 1:

    The revised paper reorganized the language expression of the conclusion. The conclusions of this paper mainly have the following two points

(1) Managerial power reduces the corporate tax avoidance, and firms with high managerial power would have lesser incentive to avoid tax.

(2) Companies need to improve managerial power through appropriate means to achieve full taxation, but also strengthen supervision and restraint on their self-interested behavior.

Point 2: The introduction and the literature review should be more consequent (it is nor so much the content issue as structure of logic it is linked).

Response 2:

In the revised paper, a new logical combing of the introduction and the literature review section is carried out.

In the introduction, firstly, the importance of taxation and the multiplicity of tax avoidance phenomenon are pointed out. Secondly, different viewpoints held by different scholars on the influencing factors of tax avoidance behavior are pointed out, which leads to the particularity and research urgency of the managerial power on corporate tax avoidance in the special market environment of China. And finally the author explains the research methods, structure and the possible contributions of this paper.

The literature review section was divided into three parts: corporate tax avoidance, managerial power, and hypothesis. In the corporate tax avoidance part, the factors affecting corporate tax avoidance are divided into two parts: impeding or promoting. The main keywords are concentrated on the “the management”, and the literature on the factors of researching managerial power is rarely appear. The managerial power section studied the impact of managerial power on financial quality, financial management, business climate, and decision-making behavior. It is found that the literature directly researching on managerial power and corporate tax avoidance behavior was almost blank. Assuming the proposed part, the author proposed the following two hypotheses

Hypothesis (H1): Managerial power would inhibit corporate tax avoidance.

Hypothesis (H2): Firms with high managerial power would have lesser incentive to avoid tax.

Point 3: The Hypotheses, I think, should be reformulated, using the same semantics, definitions to make it scientifically clear, non ambiguous.

Response 3:

    The revised paper had logically combed the supporting documents of the two hypotheses, and individually expressed the background of the two hypotheses. I hope that such expressions can gain more understanding from readers.

Point 4: The discussion at the end of the paper should be developed further to discuss the results in the light of different taxation systems, different economic levels, revealing the specificity of results.

Response 4:

    In the 4th paragraph of the Section5, the implications of this paper are explained in detail from the theoretical and practical aspects. In terms of practical implications, new public policy perspective has been introduced for readers can feel more about the value of this paper.

Point 5: I would suggest to improve conclusions, while expressing more clearly the final resolutions.

Response 5:

    The revised paper had fully discussed the statistical data and empirical results of this paper to avoid the pale conclusion.

Point 6: I'm not qualified to judge English language and style, but even for me it is clear that English language style should be improved. There many small redaction mistakes to be corrected.

Response 6:

    The new version of paper not only revised the content, but also edited and polished the English style. At the same time, I also handed the paper to professional institutions for retouching, but due to the time relationship, the paper is still being edited, and the retouched paper will be submitted in the next round of revision. I am very sorry about this. Please forgive.

Round 2

Reviewer 1 Report

Dear Authors,

here are my comments to the new version of your paper:

1) Please, check the English language, the current version needs a thorough language edit.

2) The Discussion was substantially improved, yet this section would still need certain development.

Now, authors explain the value of their results, but do not discuss their findings "with" the prior literature on management power. Perhaps it would be beneficial to frame your new results within the:

i) previous findings on management power and show clearly how your results help to develop some of the previous studies, while

ii) showing clearly also studies that are in contrast with your results on TA and MP, together with trying to explain these inconsistencies/commonalities.

Nevertheless, the new version of the paper was substantially improved. Therefore, I suggest only "minor revisions" in the second round of the review process.

Thank you.

Kind regards,

Reviewer

Author Response

Point 1: Please, check the English language, the current version needs a thorough language edit. 

Response 1: This version of the paper had been polished by professional institution.There is a touch-up certificate in the attachment.

 Point 2:The Discussion was substantially improved, yet this section would still need certain development. Now, authors explain the value of their results, but do not discuss their findings "with" the prior literature on management power. Perhaps it would be beneficial to frame your new results within the previous findings on management power and show clearly how your results help to develop some of the previous studies, while showing clearly also studies that are in contrast with your results on TA and MP, together with trying to explain these inconsistencies/commonalities. 

Response 2: In the Section5"Conclusions and Discussion", the author revised some of the expression to make them more closely related to the previous literature, and on this basis, made some contributions to the research. I hope that this expression can be more logical.

Reviewer 2 Report

The revised version has addressed the comments and suggestions on the original submission.

Author Response

Point 1: English language and style are fine/minor spell check required. Response 1: This version of the paper had been polished by professional institution.There is a touch-up certificate in the attachment.

Reviewer 3 Report

No comments.

Author Response

Point 1: I don't feel qualified to judge about the English language and style. Response 1: This version of the paper had been polished by professional institution.There is a touch-up certificate in the attachment.
